# Projecting Informal Care Demand among Older Koreans between 2020 and 2067

**DOI:** 10.3390/ijerph19116391

**Published:** 2022-05-24

**Authors:** Bo Hu, Peter Shin, Eun-jeong Han, YongJoo Rhee

**Affiliations:** 1Care Policy and Evaluation Centre, Department of Health Policy, London School of Economics and Political Science, London WC2A 2AE, UK; b.hu@lse.ac.uk; 2Milken Institute School of Public Health, Department of Health Policy and Management, George Washington University, Washington, DC 20052, USA; pshin@gwu.edu; 3Health Insurance Policy Research Institute, National Health Insurance Service, Wonju 26464, Korea; 9739han@gmail.com; 4Department of Health Sciences, Dongduk Women’s University, Seoul 02748, Korea; 5Department of Psychiatry & Behavioral Sciences, Feinberg School of Medicine, Northwestern University, Chicago, IL 60611, USA

**Keywords:** projections, informal care demand, older people, Korea

## Abstract

Background: The number of Korean older people receiving informal care is expected to rise sharply due to aging population. This study makes projections of demand for informal care in community-dwelling older people aged 65 and over in Korea until 2067. Method: The study drew on data collected from waves 4–6 of the Korean Longitudinal Study of Aging (2012–2016, *n* = 12,975). Population data published by Statistics Korea and data from the Long-term Care Insurance Statistical Yearbook for Korea were also used. A macro-simulation model was built to make the projections. Results: The number of older people receiving informal care will increase from 0.71 million in 2020 to 2.2 million in 2067. Demand for informal care from adult children or relatives is projected to rise by 257%, much faster than the increase in demand for spousal care (164%). The estimates are sensitive to alternative assumptions about future mortality rates, fertility rates, patterns of migration, and the prevalence of functional disabilities in the population. Conclusion: Demand for informal care in Korea will rise substantially in the coming decades, and the increase will be uneven for different groups of care users. Our analyses are not only relevant to the long-term care system for the general older population but also have profound implications for intensive users of long-term care in Korea. The findings highlight the importance of accurate identification of unmet needs in the population and timely delivery of government support to older people and their informal caregivers.

## 1. Introduction

South Korea’s population is aging faster than many other countries. The proportion of older people is expected to increase from 15% in 2019 to 44% in 2067, and due to persistent low birth rates and increasing life expectancy, the number of older people supported by every 100 working-age adults is expected to more than triple from 37.6 to 120.2 over this period [1]. This disproportionate change in the older and working-age population will lead to a rapid increase in demand for long-term care services and the costs of care for older Koreans. 

South Korea established the long-term care insurance (LTCI) system in 2008, providing formal and home-visit care services for older Koreans who need assistance to live independently [2]. Due to limited resources, LTCI is only accessible to those who pass a need assessment. Priority is given to older people with the highest level of needs and to a smaller proportion of younger citizens with major chronic health issues or disabilities. As of 2020, only 11.1% of elderly Koreans are able to receive long term care insurance [3,4]. As LTCI coverage slowly expands, total LTCI expenditures are expected to increase from 0.54% of GDP in 2020 to 4.2% by 2050, threatening the financial viability of long-term care coverage [5,6].Care by family members is a key factor that mitigates the economic pressure on Korea’s long term care system. Studies show informal or unpaid caregivers, combined with professional caregiver support, can help effectuate medical care [7,8,9]. It has been found that those who were more fragile and used the long-term care system continued to rely on family to act as unpaid caregivers [10]. Spouses in particular play a major role as unpaid caregivers [11], as do adult children. However, decreasing marital and birth rates are increasingly shifting the responsibilities of care provision toward the formal care system and professional caregivers [12]. Previous studies have projected that demand for informal care in the Chinese older population will double in the next decades and long-term care expenditure, as a proportion of GDP in Hong Kong, will rise from around 1.5% in 2016 to 3.0% in 2036 [13,14]. 

To date, little research has been done to understand how much the growing Korean elderly population can depend on unpaid caregivers. There is no estimate of how many informal caregivers will be needed and the potential shortfall as the number of older people continues to grow without major changes in LTCI. In this context, the present study seeks to address the following questions: What are the population-level estimates of the population who need community-based long-term care?How does informal care differ by the level of care needs, marital status, and family living arrangement?How many informal and formal caregivers will be needed by 2067?

## 2. Research Methods

### 2.1. Data

Our main source of data is the Korean Longitudinal Study of Ageing (KLoSA) study (*n* = 22,925) which used Employment Information Service data on ageing and care-related information from a nationally representative sample between 2012 and 2016 [15]. We projected population estimates using the Korean Long-term Care Insurance Statistical Yearbook 2019 and Population Projections for Korea [1,3].

### 2.2. Projection Modelling

Our modelling approach consists of two stages. In the first stage, we conducted binary logistic regression analyses using KLoSA data to identify the most important factors associated with informal and formal care. For recipients of informal care, we applied binary logistics regression analyses to identify those likely to receive care from spouses or others including children, relatives, or friends.

The selection of factors into the regression models was based on the behavioral model of care utilization which divides the determinants of care utilization into three groups: need, predisposing, and enabling factors [16,17]. Long-term care needs are measured by older people’s functional limitations. KLoSA asked respondents whether they could perform seven basic activities of daily living (ADLs; dressing, washing face/hair/brushing teeth, showering/bathing, eating, walking, using the toilet, and bladder/bow management) and ten instrumental activities of daily living (IADLs; grooming, housekeeping/bedding, preparing meals, doing laundry, walking short distances, independent transport, shopping, managing money, using the telephone, taking medication) [18]. Following Pickard, Wittenberg [19], we created an ordinal need variable with five categories: no ADL or IADL difficulties (independence), IADL difficulties only, one ADL difficulties, two ADL difficulties, and three or more ADL difficulties. We investigated age and gender as two predisposing factors. For the enabling factors, we examined marital status, living arrangements, education, and income. We combined marital status and living arrangements into one variable with four categories: single living alone, single living with others, married couples living alone, and married couples living with others. Single people include those who were never married, separated, widowed, and divorced. The education variable was dichotomised completion of primary and secondary/tertiary education. 

In the second stage, we used Care Policy and Evaluation Centre’s method (CPEC, formerly known as PSSRU) to build a projection model [20]. The same approach has been used in other regions including England, mainland China, and Hong Kong [13,14,21] and for different groups of long-term care recipients such as older people with dementia or people who have had strokes [9,22]. The model started by dividing the total population of older people by age and gender. Using data from the Long-term Care Insurance Statistical Yearbook for Korea [3], we separated the older population living in the community from those in nursing homes. For people living in the community, we divided them into 400 small groups according to the six most important factors identified in the regression analyses: age, gender, care needs, marital status, living arrangements, and education. The parameters used to divide the population were estimated using the KLoSA data. Then we estimated the predicted probability of receiving informal or formal care for each small group based on multinomial logistic regression analyses. Multiplying the predicted probability and the number of older people gives us the total number of informal care and formal care recipients in each small group. 

Next, we focused on people receiving informal care and conducted binary logistical regression analyses to estimate the probability of receiving care from the spouse or other family members according to age, marital status, living arrangements and education, as these factors are significantly associated with sources of informal care in the regression analyses. By aggregating the number of recipients in each small group, we calculated the total number of recipients of informal (including spousal and non-spousal care) and formal care at the national level in the 2020 base year. In the projection years, we plugged in the projected number of old people by age and gender reported in the official population data, which enabled us to estimate the projected demand for informal and formal care. It should be pointed out that, in reality, multiple caregivers may provide care for one person. For example, an older adult may receive both informal care and formal care or receive both spousal and non-spousal. KLoSA asked survey participants to report their main caregivers. Therefore, our logistic regression analyses of caregivers and projections of demand for informal and formal care related to main caregivers only. The base case of the projection model relies on five key assumptions:The total number of people aged 65 and over divided by age and gender changes in line with Statistics Korea’s medium growth scenario.The prevalence rates of ADL and IADL difficulties by age and gender remain constant in the projection years.Older people’s marital status and living arrangements by age, gender, and care needs remain constant.Older people’s level of education by age, gender, care needs, marital status, and living arrangements remain constant.There are no constraints on the supply of long-term care in the projection years so increases in the receipt of care reflect changes in care demand only.

A series of analyses were undertaken to the sensitivity of our projection results to the assumptions in the base case. First, we made projections based on high and low scenarios of population growth [1]. The high growth scenario assumes a high fertility rate, high life expectancy, and a high international migration level, whereas the low growth scenario assumes a low fertility rate, low life expectancy at birth, and low international migration level. Second, future prevalence rates of functional difficulties will also be impacted by factors that increase lifespans including the effectiveness of morbidity programs, care services, and efforts to live a healthier lifestyle, as well as technological advancements. Following Wittenberg, Hu [8], we investigated a scenario with increasing care needs where the prevalence of ADL and IADL functional difficulties in the older population increases by 0.5% each year until 2040 and one with care needs decreasing by 0.5% until 2067, respectively. Another Korean national study, the Korean Long-term Care Status Report [23], suggested that the sample collected in the KLoSA is healthier than the overall older population, meaning KLoSA may have underestimated the prevalence of care needs. In particular, the prevalence of IADL difficulties is 69% higher and the prevalence of ALD limitations is 58% higher, respectively, in that report than in KLoSA. To account for these uncertainties, we also used the prevalence rates reported in the Korean Long-term Care Status Report [23]. 

### 2.3. Research Findings

The data were pooled from waves four to six of the biennial KLoSA (2012, 2014, and 2016). Those who were relocated to nursing homes where formal care is provided in institutionalized settings were excluded from the study as the care in nursing homes is not comparable to caregiving in the community. Among the 12,975 respondents in our sample, 82% (*n* = 10,663) had no functional difficulties in performing ADL or IADL tasks, 10% (*n* = 1301) reported having IADL difficulties but no ADL difficulties, 3% (*n* = 362) reported having one or two ADL difficulties, and 5% (*n* = 649) reported having more than three ADL difficulties (Table 1). Among respondents who reported more than three ADL difficulties, 43.1% received care from informal caregivers while 16.2% reported having any support from formal caregivers. Those who reported one or two ADL difficulties had more support from informal caregivers (38.2%) and only 8.6% received any formal care support. It is important to reiterate here that the receipt of informal and formal care refers to care provided by main caregivers.

Columns 2 and 3 in Table 2 show the regression results in relation to predictors of long-term care service use. Compared to respondents who reported not receiving informal/formal care, participants who did so were significantly older, had lower educational attainment, and have more deficits in ADL and IADL limitations. Married people and people living with other family members are more likely than those living alone to receive informal care. No significant differences were observed between the two groups by gender or the mean number of household members. Among informal care recipients, women in higher age groups were more likely than younger men to receive informal non-spousal care as opposed to spousal care (Column 4, Table 2). The majority of non-spousal care was provided by adult children. Those with primary education or above were more likely to receive spousal care and those without formal education were more likely to receive care from children or other relatives (namely non-spousal care). 

On the basis of the regression analysis, Figure 1 and Figure 2 shows older people’s propensity to receive long-term care varies greatly according to personal characteristics. People with low care needs (IADL difficulties only) have a probability of 0.28 of receiving informal care and a probability of 0.03 of receiving formal care. For people with high care needs (three or more ADL limitations), the probability of receiving informal and formal care increased to 0.43 and 0.16, respectively (Figure 1). Among informal care recipients, men aged between 65 and 74 years old have a probability of 0.97 of receiving care from a spouse and a probability of 0.03 of receiving care from other family members. This is in stark contrast to female informal care recipients aged 85 and over who have a probability of 0.07 of receiving care from a spouse and a probability of 0.93 of receiving care from other family members (Figure 2). 

Table 3 shows the projected number of older people with care needs. According to Statistics Korea (2019), the number of older people aged 65 and over is projected to increase by 125%, from 8.1 million in 2020 to 18.3 million in 2067. The number of people aged 85 and over will rise by 564%, much faster than the older population in general. We project that the number of community-dwelling older people with long-term care needs will triple by 2067 and the prevalence of care needs will increase from 16.4% in 2020 to 23.4% in 2067. We estimate that there are 197,000 older people with one or two ADL difficulties and 353,000 older people with three or more ADL difficulties in 2020. We project that these figures will rise by 258% and 249%, respectively, by 2067. In comparison, the number of older people with a lower level of care needs, namely those with IADL limitations only, is projected to increase by 186%, from 744,000 in 2020 to 2.1 million in 2067. A faster increase in the number of people with a higher level of care needs is attributable to a rising proportion of older people in the higher age groups whose care needs are higher, as shown in Table 1 and Table 3. 

Table 4 shows the projected demand for long-term care for older people living in the community. We project that the number of people receiving informal care will rise from 0.71 million in 2020 to 1.7 million in 2040 and 2.2 million in 2067, reflecting projected changes in Korea’s demographic profile. The demand for formal care is estimated to be 130,000 people in 2020 and 388,000 people in 2067, a similar pace of increase as informal care needs. 

The increase in demand for informal care varies by group. Demand for informal care for single older people is projected to increase by approximately 250%, from 269,000 people in 2020 to 973,000 people in 2067. In comparison, care demand for informal care among married couples will increase by 180%, from 0.44 million people to 1.2 million people (Table 4). While we estimate that there were 64,000 more older people receiving spousal care than non-spousal care in 2020, this pattern will be reversed in the following decades, with 130,000 more people receiving non-spousal care than spousal care by 2067. 

Under Korea Statistics’ (2019) high population growth scenario, the number of people receiving informal care is estimated to increase from 0.71 million in 2020 to 2.4 million in 2067, a rise of 236%, compared to 206% in the base case (Table 5). Under its low population scenario, the number of informal caregivers is projected to increase by 176%, from 0.71 million in 2020 to 2.0 million in 2067. Under our increasing needs scenario, we project that the number of people receiving informal care will increase to 2.3 million by 2067, a rise of 225%, compared to 206% in the base case. Under our decreasing needs scenario, we project that demand for informal care will increase to 2.1 million people by 2067. Meanwhile, if we assume the older population has a higher prevalence of care needs in the base year, as indicated by the Korean Long-term Care Status Report, and that the prevalence remains unchanged in the projection years [23], the number of people receiving informal care will increase from 0.86 million in 2020 to 2.4 million in 2067, as opposed to 0.71 million in 2020 to 2.2 million in the base case. 

## 3. Discussion

This is the first study to estimate demand for unpaid informal caregivers for older Koreans and to identify critical implications for the long-term health care workforce policymaking. Drawing on data from the national survey of KLoSA, this study investigated the factors associated with the use of long-term care for older people and made projections of the demand for informal and formal home care. Driven by an ageing population, demand for long-term care among community-dwelling older people will increase tremendously. It has been projected that demand for informal care in the Chinese older population will rise by around 100% between 2015 and 2035. We have projected in this study that informal care demand will rise by around 150% between 2020 and 2040 (from 0.7 million people to 1.7 million people) [13]. To keep up with the demographic change, the number of people receiving informal and formal home care is projected to triple by 2067. 

Moreover, the increase in care demand will be uneven in the older population. We projected that the increase in care demand will be large among older people with severe functional difficulties or single older people. This is mainly driven by the fact that people in higher age groups had more severe functional difficulties and were more likely to lose spousal caregiving due to widowhood. As population ageing continues, the proportion of people in older age groups (e.g., people aged 85+) will rise, resulting in a higher prevalence of severe functional difficulties and widowhood in the older population. Approximately one in two caregivers is spousal caregiver and our projections suggest their role is likely to remain critical over the next four decades. In the absence of spousal care, children and other family members will face increasing responsibilities, which will lead to an increase in non-spousal care. Almost all females aged 85+ receive non-spousal care. We project that an increase in demand for non-spousal care will be 1.6 times greater than that for spousal care (i.e., 257%/164%). Strengthened support from the state and the third sector for non-spousal caregivers, especially adult children, will be crucial because, in addition to providing care for older family members, they often need to fulfill other responsibilities in life such as raising babies or want to pursue their career goals. 

This study confirms that the size of the caregiving workforce will have to increase substantially. The study projects a three-fold increase in the need for both formal and informal caregivers. As informal caregiving is an adequate substitute for formal caregiving, it is unclear how this model can be sustained or better supported. Due to higher levels of education and entries into the job market, grown-up children are less likely to serve as primary caregivers, which may shift caregiving responsibilities to a societal support system [24]. These findings raise several policy questions, including how to expand the formal caregiving workforce and LTCI, and find alternatives for older people’s care. In addition, the demand pressure in long-term care services will be especially concerning for people with dementia or stroke, because they are among the most intensive users of formal and informal care [11,24]. Arguably, the same can be said about people with other chronic diseases that cause frailty such as heart failure or renal disease. Future research will benefit from the projections of long-term care demand among people with those diseases. 

The issue of care availability is further complicated by the recent trends in living arrangements in Korean society. Single older people or older couples without children have increased to 78.2% in 2020 compared to that of 66.8% in 2008, whereas older Koreans living with children decrease to 20.1% in 2020 compared to that of 27.6% in 2008 [24,25]. Older adults without any co-residing family members are more likely to have unmet care needs [26,27] or need to relocate to long-term care facilities [7]. Unless the government steps up its effort to support both older people with care needs and their caregivers, adequacy of care and ageing in place, which are essential to the well-being and quality of life of older people, are likely to be greatly compromised. 

Following the previous studies, we assume in the base case that the prevalence of ADL and IADL care needs by age and gender will remain constant by 2067. We project that, if the prevalence of ADL and IADL care needs instead rises by 0.5 percent each year, demand for informal care will be 0.1 million people higher by 2067 in comparison to the base case. Such a finding highlights the important role that healthy-ageing policies can play in managing future demand for informal care. Programs and prevention measures that aim to promote healthy ageing not only reduce morbidity and care needs in the older population, but also help to alleviate the mounting demand for long-term care. 

Our projections are not only relevant to policymaking and practice for older people’s care in general, but also likely to have important implications for different sub-groups of care recipients who are among the most intensive users of long-term care. Other countries have sought to estimate demand for long term health care services by older population in East Asia, such as China and Hong Kong [20,21]. Like Korea, these countries are facing very similar patterns in population component changes with increasing life expectancy and lower fertility rates and are seeking greater insights on health care service demands by a growing aging population. 

It is also important that the government take action to ensure adequate support for informal caregivers. A variety of services are available to support informal caregivers in European countries such as counselling services, financial support, and respite care [28,29]. Korea has made some effort to embrace caregiving in the system for more support among older Koreans and their family members. Since 2017, for example, Korea has been developing a “National Responsibility System for Dementia” in the community, which includes education and counselling services and financial support for family caregivers supported by local government programs and LTCI [30]. Moreover, family caregivers having high burden and co-residing with older adults who qualify for LTCI are eligible for COMPASS program, which is multi-component program including counselling, education, support group services and art therapy [31,32,33]. However, the extensive family support program of local government is limited to primary informal caregivers caring for patients diagnosed of dementia by the health insurance system, leaving most family caregivers without access to this type of benefit. The COMPASS program of LTCI is also not yet widely available for all primary family caregivers for LTCI recipients. In order to adequately meet the growing demand for caregivers, such limits and barriers to educational and financial supports will need to be addressed. 

## 4. Limitations

As with any projected estimates, there are great uncertainties in relation to demographic and epidemiological trends, such as, fertility rates, mortality rates, and prevalence of functional disabilities in the population. Our analyses of the KLoSA data show that there were no substantial changes in the prevalence of functional disabilities in Korean older population in the past few years. Therefore, it seems plausible to assume that the prevalence is constant in the base case projections and our results will be useful approximations of the care demand in the short term (i.e., next few years). The same can be said about the assumption on fertility and mortality rates [1,5]. Moreover, we have tried to capture these uncertainties by conducting sensitivity analysis. More focused studies on those trends will also be valuable to reduce the uncertainties in making projections, which will be valuable to policymakers and practitioners. Our estimates are based on KLoSA survey which tends to reflect younger and healthier older adults than the general elderly population and with low prevalence rates in severe illnesses such as heart failure or chronic renal disease. Due to this limitation, we did not consider making disease estimates [30,34]. As a result, our estimates for caregivers may be considered conservative. Finally, this study excluded the older adults in nursing homes who receive formal care, which poses another source of underestimation of demand for long-term care. 

## 5. Conclusions

The demand for informal care for older Koreans is likely to increase three-fold by 2067. We anticipate that the demand for informal care will keep rising as the proportion of older people in higher age groups and the prevalence of care needs continue to rise. There will be a greater increase in demand for care from adult children than that for spousal care due to the increased prevalence of widowhood. Yet, adult children may take less of a role in caregiving as education levels and female employment rise and the proportion of single older people living apart from children gradually increases. Timely identification of older people with unmet needs and delivery of high-quality support to care recipients and their caregivers will be the key to addressing the mounting demand for informal care in Korean society. 

## Figures and Tables

**Figure 1 ijerph-19-06391-f001:**
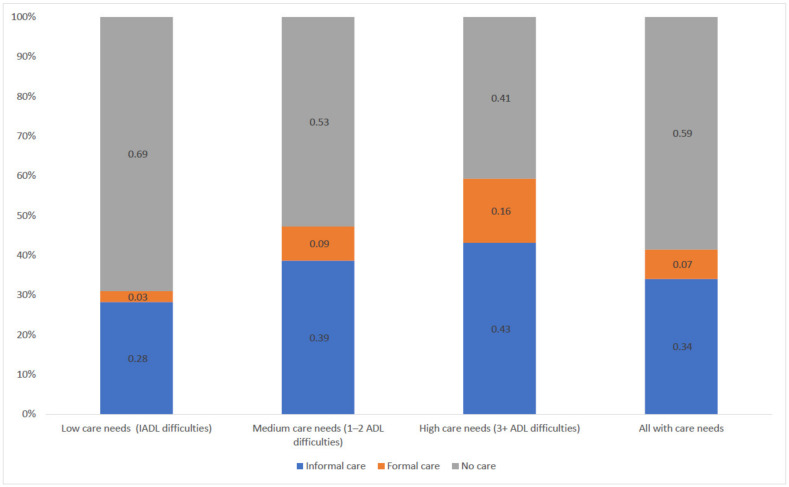
Predicted probability of receiving informal or formal care by levels of care need.

**Figure 2 ijerph-19-06391-f002:**
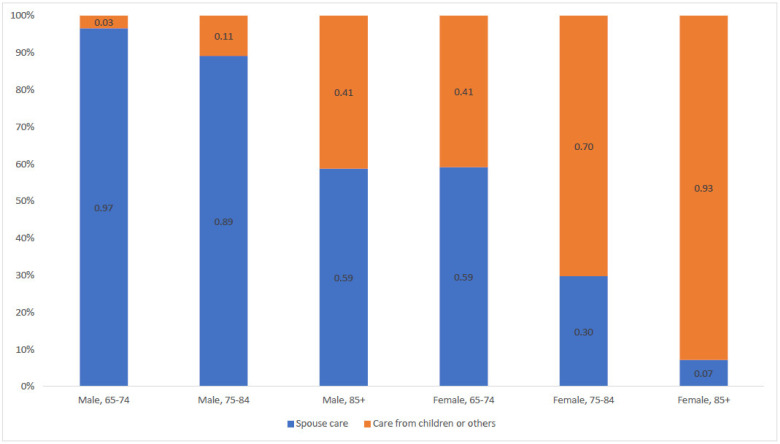
Predicted probability of receiving care from spouse or other family members by demographic profiles among informal care recipients.

**Table 1 ijerph-19-06391-t001:** Sample Characteristics.

		Level of Functional Capability *n* = 12,975			
		Independent *n* = 10,603	IADL Difficulties Only *n* = 1301	1–2 ADL Difficulties *n* = 362	3+ ADL Limitations *n* = 649
		%	%	%	%
Age	65 ≤ Age < 70	28.45	14.99	6.91	6.01
	70 ≤ Age < 75	27.78	15.76	12.43	13.10
	75 ≤ Age < 80	23.62	23.98	18.51	19.72
	80 ≤ Age < 85	13.59	24.21	22.65	25.73
	85 ≤ Age	6.56	21.06	39.50	35.44
Age (Means, Std)		74.04 (6.29)	78.36 (7.33)	81.84 (7.89)	81.85 (7.86)
Gender	Female	58.19	50.19	67.13	61.33
Marital Status	Married	68.97	61.72	45.30	53.62
	Single/Separated/Widow	31.03	38.28	54.70	46.38
Living arrangement	Living alone	14.53	13.45	14.09	13.41
	Single Living with others	16.51	24.82	40.61	32.97
	Married couple living alone	17.35	14.53	11.05	13.71
	Married couple living with others	51.62	47.19	34.25	39.91
Education	Elementary school	59.19	64.95	81.77	78.58
	Middle school	15.51	12.76	8.01	7.70
	High school	18.33	15.60	7.18	10.63
	More than college	10.63	6.69	3.04	3.08
ADL (mean, std)		NA	NA	1.31 (0.47)	5.95 (1.50)
IADL (mean, std)		NA	3.31 (2.22)	5.80 (2.71)	9.18 (1.66)
Care receipt	Not receiving care	95.04	69.02	52.76	40.68
	Informal care (main carer)	4.38	28.21	38.67	43.14
	Formal care (main carer)	0.58	2.77	8.56	16.18

**Table 2 ijerph-19-06391-t002:** Predictors of long-term care receipt and sources of informal care.

	Multinomial Logistic Regression		Binary Logistic Regression
	Informal Care	Formal Care	Sources of Informal Care
	RRR (Standard Error)		OR (Standard Error)
65–74 years old (ref. category)			
75–84 years old	1.62 *** (0.13)	1.66 ** (0.3)	2.89 *** (0.57)
85+ years old	2.38 *** (0.25)	1.41 (0.33)	15.08 *** (3.77)
Male (refence category)			
Female	0.86 (0.07)	1.41 (0.26)	15.14 *** (2.73)
Single living alone (ref. category)			
Single living with others	1.88 *** (0.23)	0.79 (0.17)	N.A.
Married couples living alone	0.99 (0.15)	0.42 ** (0.13)	N.A.
Married couples living with others	2.03 *** (0.25)	1.23 (0.26)	N.A.
Independent (ref. category)			
IADL difficulties only	7.44 *** (0.6)	6.11 *** (1.33)	1.20 (0.23)
1–2 ADL difficulties	11.86 *** (1.5)	22.44 *** (5.42)	0.90 (0.25)
3+ ADL difficulties	17.81 *** (1.83)	57.29 *** (10.46)	0.76 (0.17)
No formal education (ref. category)			
Primary education or above	0.82 * (0.07)	0.54 ** (0.11)	0.33 *** (0.07)

Note: For multinomial logistic regression, the base outcome is no care; For binary logistic regression, 0 = spouse care and 1 = receiving care from children or other family members. * *p* < 0.05, ** *p* < 0.01, *** *p* < 0.001

**Table 3 ijerph-19-06391-t003:** Projected population and care needs (unit: thousand people).

	2020	2030	2045	2067	% 2020–2067
Demographic trends					
Older people 65+	8125	12,980	18,329	18,271	125%
Older people 85+	772	1444	3257	5124	564%
People with care needs in the community					
Independent	6617	10,541	14,078	13,330	101%
IADL difficulties only	744	1194	1946	2128	186%
1–2 ADL difficulties	197	322	582	707	258%
3+ ADL difficulties	353	573	1036	1229	249%
People with care needs	1294	2089	3564	4064	214%
Prevalence of care needs	16.4%	16.5%	20.2%	23.4%	43%

**Table 4 ijerph-19-06391-t004:** Projected demand for community-based long-term care (unit: thousand people).

	2020	2030	2045	2067	% 2020–2067
Main caregivers					
Informal care	706	1139	1908	2162	206%
Formal care	130	205	350	388	199%
Living arrangements					
Single living alone	68	108	197	239	251%
Single living with others	201	319	596	734	266%
Married couples living alone	70	114	182	198	183%
Married couples living with others	367	598	932	992	170%
Sources of informal care					
Spouse	385	626	964	1016	164%
Non-spouse only	321	512	944	1146	257%

**Table 5 ijerph-19-06391-t005:** Sensitivity analysis of projected demand for informal care (unit: thousand people).

	2020	2030	2045	2067	% 2020–2067
Base case	706	1139	1908	2162	206%
Population scenarios					
High population growth	706	1164	2008	2375	236%
Low population growth	706	1111	1801	1951	176%
Care needs scenarios					
Increasing care needs	706	1169	2019	2295	225%
Decreasing care needs	706	1140	1906	2070	193%
Caregiving study	858	1354	2217	2422	243%

## Data Availability

All authors have full access to all the study data.

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
