# Peer review of "Projecting Informal Care Demand among Older Koreans between 2020 and 2067"

_ijerph, 2022, doi:10.3390/ijerph19116391_

Round 1

Reviewer 1 Report

The authors have sent us a very interesting work about the prevision of informal care in older patients. In my opinion the work is correct, is well written and gives response to a important problem in the occidental countries. But, for be published in the journal, maybe they have to do some changes.

Abstract

  • The authors conclude that their results have profound implications for intensive users of long-term care in Korea such as people living with dementia. But there are any results about this in the results point. Certainly, we can expect this situation, but I think is better to express in other form, for example: with these results, we can expect the situation will be worst in intensive patients of long term care in Korea such as people living with dementia and other disease that cause frailty in the older.

Methods

  • In the abstract and in the discussion, authors explain some opinion about patients with dementia. It is possible to think the relationship between their conclusions and patients with theses disease, but they have to express if they had any analyses using patients with this disease. If yes, express also results, if not, please delete the comment about this disease.

Results

  • The authors must indicate the units in tables 3, 4 and 5.

Discussion

  • The authors express in 6th paragraph that their results are important in disease like dementia, but I did not understand this sentence because there are not any results about theses disease. And other question about this is why the author do not analyse other disease as heart failure or chronic renal disease that also causes fragility.
  • In the limitations, authors indicate that the statistics models that they used in this work do not include fertility rates, mortality rates, and prevalence of functional disabilities in the population; but at least, in my opinion, it is possible to analyse the rates of the last years of theses three situations, and if they are stable, we can accept that the data that the authors show will be correct in the next years.

References:

  • The authors use 11 references older than 5 years. I think that some of them will be changed by other more recent.

Author Response

Reviewer 1

Comments and Suggestions for Authors

The authors have sent us a very interesting work about the prevision of informal care in older patients. In my opinion the work is correct, is well written and gives response to a important problem in the occidental countries. But, for be published in the journal, maybe they have to do some changes.

Response

Thanks for detail comments to improve our manuscript. We changed the manuscript according to comments and highlighted them in the main manuscript so you can check them.

1). Abstract

The authors conclude that their results have profound implications for intensive users of long-term care in Korea such as people living with dementia. But there are any results about this in the results point. Certainly, we can expect this situation, but I think is better to express in other form, for example: with these results, we can expect the situation will be worst in intensive patients of long term care in Korea such as people living with dementia and other disease that cause frailty in the older Koreans.

2) Methods

In the abstract and in the discussion, authors explain some opinion about patients with dementia. It is possible to think the relationship between their conclusions and patients with theses disease, but they have to express if they had any analyses using patients with this disease. If yes, express also results, if not, please delete the comment about this disease.

Response

We thank the reviewer this comment.  Following this comment, we have expanded the third paragraph in the discussion section to (1) incorporate the reviewer’s point that the situation will be even more serious for people living with dementia and other disease; (2) clarify that we did not have data to make projections for people with chronic disease such as dementia but future research that can make projections of LTC among people with those diseases will be especially helpful.

3). Results

The authors must indicate the units in tables 3, 4 and 5.

Response

Thanks for pointing this out. We have clarified that the unit is thousand people in those tables in the revised manuscript.

Change

Table 3. Projected population and care needs (unit: thousand people

Table 4. Projected demand for community-based long-term care (unit: thousand people).

Table 5. Sensitivity analysis of projected demand for informal care (unit: thousand people)

4). Discussion

The authors express in 6th paragraph that their results are important in disease like dementia, but I did not understand this sentence because there are not any results about theses disease. And other question about this is why the author do not analyse other disease as heart failure or chronic renal disease that also causes fragility.

Response

Thanks for your insightful input on disease issue. The reviewer is correct that we did not have reliable data on dementia and so made no such projections.  We noted in the initial submission that the KLoSa cohort is relatively healthy and younger older Koreans who living in community, and so disease projections may be underestimated. We also had noted that  KLoSa did not include nursing home residents who would have given us more reliable projections on those who may be more physically and mentally fragile. We added this issue in limitation using new reference (34).

Change

We revised the sentence in line 320-322 as below.

Our estimates are based on KLoSA survey which tends to reflect younger and healthier older adults than the general elderly population and with low prevalence rates in severe illnesses such as heart failure or chronic renal disease. Due to this limitation, we did not consider making disease estimates [30, 34].

  1. Jeon, B.; et, al.,. The long-term care utilization of the elderly with dementia, stroke, and multimorbidity in Korea. Health Policy & Management, 2013, 23, 90-100.

For the 6th paragraph, we highlight dementia as an example of resources available to informal givers to those with dementia

Korea has made some effort to embrace caregiving in the system for more support among older Koreans and their family members  Since 2017, for example, Korea has been developing a “National Responsibility System for Dementia” in the community, which includes education and counselling services and financial support for family caregivers supported by local government programs and LTCI [29].

In the limitations, authors indicate that the statistics models that they used in this work do not include fertility rates, mortality rates, and prevalence of functional disabilities in the population; but at least, in my opinion, it is possible to analyse the rates of the last years of theses three situations, and if they are stable, we can accept that the data that the authors show will be correct in the next years.

Response

We thank the reviewer for raising this very good point.

Change

We have incorporated this point in the limitations paragraph.5). References:

The authors use 11 references older than 5 years. I think that some of them will be changed by other more recent.

Response

We changed the reference 11 to the newer study as below.   Some of references are cited as they are relevant to method or other countries’ situation which does not have newer study so far. We updated a few references to the recent ones as possible.

Change

  1. Han, E.-J.; et al. Comparison of caregiving burdens among family members by the type of benefits in long-term care. Korea Social Policy Review, 2019, 26(3), 93-116.

Reviewer 2 Report

For the conclusions, the first sentence describes the "faster speed." Yet, the increasing age of the population is not a function of time.  It may be a function of the current age of the population and the characteristics associated with the aging of the population.  I believe the time is fixed.

The authors also reference "faster" in the discussion section.  Is it faster or a function of widowhood and other characteristics that lend themselves to the need for increased formal and informal care?

Page 3, line 130.  the "T" is missing from The

There are minor typographical and grammar.

Author Response

Reviewer 2

Comments and Suggestions for Authors

For the conclusions, the first sentence describes the "faster speed." Yet, the increasing age of the population is not a function of time.  It may be a function of the current age of the population and the characteristics associated with the aging of the population.  I believe the time is fixed.

Response

Thank you for pointing this out. Yes, we also meant that increase age is a function of current age and the characteristics, and we did not mean that it is a function of time. We apologize for the confusion. To avoid further confusion, we have deleted the word ‘faster speed’ and re-written the concluding paragraph.

Change

The demand for informal care for older Koreans is likely to increase three-fold by 2067.

1). The authors also reference "faster" in the discussion section.  Is it faster or a function of widowhood and other characteristics that lend themselves to the need for increased formal and informal care?

Response

Thanks for your insightful comments. To answer the review’s question, yes, it is a function of widowhood and prevalence of functional difficulties in the older population aged 65 and over. Following this comment, we have deleted the sentence about faster increase in care demand. We clarified in the revised manuscript that the increase in care demand will be larger among people with ADL care needs (an increase by around 250%) than those with IADL care needs (an increase by around 185%). The increase in care demand will be larger among single people (an increase by around 250%) than married older people (an increase by around 180%).   

We have also re-written the second paragraph of the discussion section to clarify this point. We have pointed out in the revised manuscript that, as the Korean population is growing older, the number and proportion of widowed female who are older than 65 years old is also expected to increase. This is the main driver of the large increase in demand for non-spousal care.

We revised the sentences in discussion to clarify the issue. Please see the revised one as below.

Change

We revised the sentence to give more explanation about the faster growing demand of care as below.

This is mainly driven by the fact that people in higher age groups had more severe functional difficulties and were more likely to lose spousal caregiving due to widowhood.

As the Korean population continues to age, a larger proportion of people will be in higher age groups. We anticipate that the demand for informal care will increase as the proportion of older people in higher age groups and the prevalence of care needs continue to rise. Demand for care from adult children may increase due to the increased prevalence of widowhood.

2). Page 3, line 130.  the "T" is missing from The

Response

Thank you for pointing out the typo. We corrected it.

Change

Please see the revised the sentence as below. Line 130

The total number of people aged 65 and over divided by age and gender changes in line with Statistics Korea’s medium growth scenario.

3). There are minor typographical and grammar.

Response

Thank you. We reviewed the manuscript carefully and corrected all typos and awkward sentences.

Reviewer 3 Report

This paper touches upon an emergent issue and the analysis is timely regarding the rapidly changing eldercare sector in Asia nowadays. The paper is overall well written and the methods and findings are appropriately presented. I would like to ask the authors to consider some revisions before this paper is ready for publication. Please see below for my comments:

1). The research question concerns Korea, and relevant research on aging and eldercare in Korea has been presented. However, in the introduction, it would be beneficial to see what are other studies and their results in the East Asian context just to put this research in the larger picture. This would also benefit the Discussion part interpreting the possible causes and solutions for the informal sector. I would like to ask the authors to also provide some information on the significance of this paper - how is it contributing to the study of eldercare in East Asia compared to other studies? What gaps does this paper fill? What prior assumptions does this paper counter?

2). I am particularly interested in the group-level variation in the informal care sector. The authors have described the results yet I was a bit disappointed to see no in-depth discussion on what are the causes for such variation, and what are the potential policy recommendations especially since the authors have highlighted the policy aspect of their research. Combining 1) and 2), I hope to see the authors provide relevant studies in the past/other countries and how other scholars have tended to respond, and that the authors of this paper tend to respond considering the Korean context. By doing this, the significance of this research could be further highlighted. 

3). typo "he" in line 130. Though no other typos were observed, I'd like to ask the authors to double-check.

Author Response

Reviewer 3

Comments and Suggestions for Authors

This paper touches upon an emergent issue and the analysis is timely regarding the rapidly changing eldercare sector in Asia nowadays. The paper is overall well written and the methods and findings are appropriately presented. I would like to ask the authors to consider some revisions before this paper is ready for publication. Please see below for my comments:

Response

Thanks for detail inputs to improve our manuscript. We changed the manuscript according to comments and highlighted them in the main manuscript so you can check them.

1). The research question concerns Korea, and relevant research on aging and eldercare in Korea has been presented. However, in the introduction, it would be beneficial to see what are other studies and their results in the East Asian context just to put this research in the larger picture. This would also benefit the Discussion part interpreting the possible causes and solutions for the informal sector. I would like to ask the authors to also provide some information on the significance of this paper - how is it contributing to the study of eldercare in East Asia compared to other studies? What gaps does this paper fill? What prior assumptions does this paper counter?

Response

We appreciated this input. Following this comment, we have added a discussion of other studies and their projection results in the East Asian context in the third paragraph of the introduction section. In the discussion, we have also discussed the projection results in Korea in reference to findings reported in other countries (e.g. China) to interpret our results in the broader context. To highlight the significance of this study, we have added in the discussion section that this is the first study to estimate “informal family care demand” projection using Korean older population. In comparing to other similar East Asian countries, Korean demand seems to be increasing faster and greater in a short time period. This finding may provide better picture for other Asian countries which are following similar patterns of aging society such as low fertility and long life expediency. Korea has projected mainly for “formal care” demand paid by long-term care insurance, and relatively for a shorter term (Kim, 2018). Our study, in contrast, projected informal care demand considering the whole population component changes for a longer period.

Change

We added a sentence in discussion as below.

This is the first study to estimate demand for unpaid informal caregivers for older Koreans and identifies critical implications for health care workforce policymaking.  

2). I am particularly interested in the group-level variation in the informal care sector. The authors have described the results yet I was a bit disappointed to see no in-depth discussion on what are the causes for such variation, and what are the potential policy recommendations especially since the authors have highlighted the policy aspect of their research. Combining 1) and 2), I hope to see the authors provide relevant studies in the past/other countries and how other scholars have tended to respond, and that the authors of this paper tend to respond considering the Korean context. By doing this, the significance of this research could be further highlighted.

Response

Thank you for these very good comments. Following these suggestions, we have re-written the second paragraph to explain causes for such variation. We have also expanded that paragraph to set out the potential policy measures that could be taken.   To highlight the contribution of this research, we have added in the discussion section that this study provides the first estimate of caregiving needs using simpler survey data but those estimates represents critical first insights for health care policymaking on caregiving needs.  However, to better highlight the need for this kind of study on Korea’s populations, we noted other regional countries that have sought similar insights.

Change

We added the following sentences in discussion.

Other countries have sought to estimate demand for long term health care services by older populations in East Asia, such as China and Hong Kong [20-21]. Like Korea, these countries are facing very similar patterns in population component changes with increasing life expectancy and lower fertility rates and are seeking greater insights on health care service demands by a growing aging population.

3). typo "he" in line 130. Though no other typos were observed, I'd like to ask the authors to double-check.

Response

Thank you for pointing out the typo. We reviewed the manuscript and corrected other typos.

Change

Please see the revised the sentence as below.

1           The total number of people aged 65 and over divided by age and gender changes in line with Statistics Korea’s medium growth scenario. (Line 130).

Round 2

Reviewer 1 Report

The authors have modified and answered mi recommendations perfectly. In my opinion, it is a acceptable manuscript.

Reviewer 3 Report

The authors have made sufficient revisions, the manuscript is ready to be published.